# Extracellular Vesicle Membrane-Associated Proteins: Emerging Roles in Tumor Angiogenesis and Anti-Angiogenesis Therapy Resistance

**DOI:** 10.3390/ijms21155418

**Published:** 2020-07-30

**Authors:** Song Yi Ko, Honami Naora

**Affiliations:** Department of Molecular and Cellular Oncology, University of Texas MD Anderson Cancer Center, Houston, TX 77030, USA; sko@mdanderson.org

**Keywords:** extracellular vesicles, protein sorting, membrane proteins, cytokines, tumor angiogenesis, anti-angiogenic therapy, therapy resistance

## Abstract

The tumor vasculature is essential for tumor growth and metastasis, and is a prime target of several anti-cancer agents. Increasing evidence indicates that tumor angiogenesis is stimulated by extracellular vesicles (EVs) that are secreted or shed by cancer cells. These EVs encapsulate a variety of biomolecules with angiogenic properties, and have been largely thought to stimulate vessel formation by transferring this luminal cargo into endothelial cells. However, recent studies have revealed that EVs can also signal to recipient cells via proteins on the vesicular surface. This review discusses and integrates emerging insights into the diverse mechanisms by which proteins associate with the EV membrane, the biological functions of EV membrane-associated proteins in tumor angiogenesis, and the clinical significance of these proteins in anti-angiogenic therapy.

## 1. Introduction

Tumor growth and metastasis depend on the development of a tumor vasculature to supply oxygen and nutrients. It is widely recognized that tumors secrete a repertoire of soluble factors that stimulate recruitment of endothelial progenitors, growth and maturation of endothelial cells, and assembly of these cells into vessels [1]. In the past decade, there has been increasing evidence that communication between cancer cells, endothelial cells, and other types of stromal cells is not only mediated by soluble factors and direct cell-to-cell contact, but also by extracellular vesicles (EVs) [2,3,4]. EVs are membranous structures that encapsulate a variety of biomolecular cargo such as DNA, RNA, proteins, and lipids and are broadly classified into the following three types based on their size and biogenesis [5,6,7]. Apoptotic bodies are typically 1 to 5 μm in diameter and are formed from blebbing of the plasma membrane in cells undergoing apoptosis. Ectosomes (also known as microvesicles or microparticles) range from 100 to 1000 nm in diameter, are formed through outward budding of the plasma membrane, and then pinch off from the cell surface (Figure 1). Exosomes range from 30 to 150 nm in diameter, derive from late endosomal structures called multivesicular bodies (MVBs), and are released into the extracellular milieu upon fusion of MVBs with the plasma membrane (Figure 1). Because of the difficulty in defining the sub-cellular origin of an EV and the lack of consensus on specific markers of the EV types, a size-based nomenclature for EVs has been recommended [8]. In this review, the original nomenclature used by each cited article will be followed.

EVs are produced by virtually all types of cells, but are often more highly secreted by cancer cells than by normal cells [2]. To date, the biological responses induced by EVs in the tumor microenvironment and in other contexts have been largely thought to be mediated through the transfer of their luminal cargo into recipient cells. A number of studies have shown that cancer cell-derived EVs stimulate endothelial cell survival, migration and tube formation in vitro and tumor angiogenesis in vivo, and have attributed these responses to various luminal constituents, such as enzymes and non-coding RNAs, that are encapsulated in EVs and transferred into endothelial cells [9,10,11,12,13]. However, recent studies have revealed that the EV membrane is not solely composed of structural components but displays a rich array of biologically active proteins with angiogenic properties. This review will not exhaustively catalog EV membrane-associated proteins, but will discuss several key examples to highlight the diverse mechanisms by which these proteins associate with the EV membrane, stimulate the angiogenic cascade, and impact on responses to anti-angiogenic therapy.

## 2. Sorting of Cell Surface Receptors and Other Integral Membrane Proteins to EVs

Mass spectrometry-based proteomic analyses of EVs derived from diverse cell types and body fluids have revealed several thousand vesicular proteins [14,15]. Of the 100 most identified EV proteins in the Exocarta database, more than 40% contain at least one transmembrane domain [16]. In an analysis of proteomic data collected from the Vesiclepedia database, 627 of 3027 evaluable EV proteins (21%) were classified as plasma membrane proteins [17]. Commonly identified EV membrane proteins include heparan sulfate proteoglycans (HSPGs) such as syndecans, tetraspanins, membrane proteases, and adhesion molecules such as integrins [16,17]. Although it is recognized that the proteome of EVs is distinct from the cellular profile, the variation in membrane protein composition between different types of EVs is poorly understood. Because ectosomes directly derive from the plasma membrane, it is conceivable that the membrane protein composition of ectosomes is more reflective of the repertoire of their parental cells, whereas the membrane protein composition of endosomal-derived exosomes is more divergent. This divergence likely stems from various sorting mechanisms as described below.

### 2.1. The Endosomal Sorting Complex Required for Transport (ESCRT) Machinery

The MVB pathway plays an essential role in regulating the sorting of integral membrane proteins, and intimately ties protein sorting to exosome biogenesis. The epidermal growth factor receptor (EGFR) is a prototypic cargo that is sorted by the MVB pathway [18]. Following ubiquitination and internalization by endocytosis, membrane proteins are routed to early endosomes. From there, the proteins either undergo recycling to the plasma membrane or are sorted into intraluminal vesicles (ILVs) that form through inward budding of endosomes and give rise to MVBs [19,20] (Figure 1). Subsequent fusion of MVBs with lysosomes results in the degradation of ILVs and their cargo by lysosomal hydrolases (Figure 1). By contrast, fusion of MVBs with the plasma membrane results in the release of ILVs as exosomes (Figure 1). The ESCRT machinery comprises four multi-subunit protein complexes (ESCRT-0, -I, -II, -III) plus the ATPase VPS4 and several accessory components that act cooperatively to orchestrate the MVB sorting pathway. Components of the ESCRT complexes and the dynamics of their assembly are described in several review articles [21,22,23]. The early ESCRT complexes (ESCRT-0, -I, -II) contain ubiquitin-binding modules. The ESCRT-0 complex serves in the initial recognition and capture of ubiquitinated cargo at the endosomal membrane, and recruits the ESCRT-I complex which confines and passes the cargo to the ESCRT-II complex [24,25]. In turn, the ESCRT-II complex serves as a scaffold for the formation of the ESCRT-III complex [26,27]. The ESCRT-III complex is the business end of the ESCRT machinery and coordinates several crucial functions. These include recruiting enzymes that remove ubiquitin moieties from the cargo, mediating the budding of ILVs, and recruiting VPS4 and other machinery to complete MVB biogenesis and disassemble the ESCRT complexes [25,28,29,30]. Intriguingly, VPS4 not only facilitates MVB biogenesis but also the formation of ectosomes [31,32].

Whereas earlier studies showed that ubiquitination of cargo is important for their recognition by the early ESCRT complexes [24,25], subsequent in silico analysis of the MVB pathway [33] and biochemical studies of specific membrane proteins have revealed that ubiquitination might not be obligatory for sorting. Protease-activated receptor 1 (PAR1) bypasses the requirement for ubiquitination and the ubiquitin-binding early ESCRT complexes by binding the ESCRT accessory protein ALG-2-interacting protein X (ALIX) that recruits the ESCRT-III complex to sort PAR1 into ILVs [34]. Tetraspanins are also sorted to exosomes via ALIX-dependent ESCRT-III recruitment [35]. Furthermore, ALIX mediates sorting of syndecans to exosomes via its interaction with the syndecan adaptor syntenin [36]. The specific cues that dictate whether MVBs fuse to lysosomes (which results in cargo degradation) or to the plasma membrane (which results in cargo being released in exosomes) have not been precisely defined, but several modifications of the cargo have been implicated. Differences in glycan epitopes in exosomes and their parental cell membranes have implicated a role for glycosylation in sorting of cargo to exosomes [37]. Sorting of specific proteins into exosomes can also be directed by SUMOylation [38] and inhibited by acetylation [39]. On the other hand, the presence of ubiquitinated proteins in exosomes has suggested that subsequent deubiquitination of cargo is not obligatory for their packaging in exosomes [40,41].

### 2.2. ESCRT-Independent Sorting

Cells that are depleted of key components of all four ESCRT complexes have been found to form MVBs, suggesting that ESCRT-independent mechanisms also regulate the MVB pathway [42]. Kajimoto and colleagues found that sphingosine 1-phosphate (S1P), a sphingolipid metabolite, controls the sorting of cargo into exosomes through inhibitory G protein-coupled S1P receptors that are located on MVBs [43]. Impairment of S1P signaling decreased the content of cargo in exosomes but not the total number or size of exosomes, indicating that S1P signaling is mainly involved in the sorting of cargo into exosomes and not in ILV formation [43]. Tetraspanins are a family of proteins that contain four transmembrane domains, interspersed with two extracellular loops and intracellular N- and C- terminal tails [44]. By engaging with one another, other membrane proteins and lipids, tetraspanins act as scaffolds to organize the plasma membrane into platforms, termed tetraspanin-enriched microdomains (TEM), which in turn facilitate transduction of extracellular signals into intracellular signaling cascades [44]. Several tetraspanins such as CD9, CD63, CD81 and CD82 are enriched in exosomes, and their ability to cluster proteins is thought to mediate the sorting of cargo into exosomes independently of the ESCRT machinery. Sorting of a melanocyte-specific glycoprotein PMEL to ILVs is mediated by its interaction with CD63 [45]. Localization of the membrane metalloprotease CD10 in exosomes is mediated by its interaction with CD9 [46]. The export of β-catenin from cells in exosomes is facilitated by CD9 and CD82 via interactions with E-cadherin [47]. Perez-Hernandez and colleagues characterized the intracellular TEM interactome by using peptides spanning the C-terminal tails of tetraspanins and tetraspanin-associated receptors as bait [48]. These authors found that the TEM interactome of CD81 and its partner EWI-2 accounted for nearly 50% of the total protein composition of exosomes [48]. Tetraspanins and their binding partners might therefore sequester a broad repertoire of membrane and cytosolic proteins into exosomes.

## 3. Sorting of Receptor Ligands to EVs

In addition to cell surface receptors, a number of ligands have been detected in EVs isolated from body fluids and from diverse cell types including cancer cells. These ligands include interleukins (IL), chemokines and members of the epidermal growth factor (EGF), vascular endothelial growth factor (VEGF), fibroblast growth factor (FGF), platelet-derived growth factor (PDGF), and transforming growth factor (TGF) families [9,49,50,51]. Because preparations of EVs that are isolated by ultracentrifugation methods can be contaminated with soluble proteins, the analysis of cytokines and other ligands in EVs necessitates more stringent methods to purify EVs (e.g., ultrafiltration and density gradient centrifugation) and rigorous validation of their localization in EVs [8]. This section discusses emerging insights into the mechanisms by which cytokines can associate with EVs and, in particular, with the EV membrane.

### 3.1. Free Secretion Versus EV Association

Following synthesis, most cytokines are trafficked to the endoplasmic reticulum (ER) and then to the Golgi apparatus, a process that is directed by their N-terminal signal peptide sequence [52] (Figure 1). Upon sorting in the trans-Golgi network, cytokines are either directed to endosomes for eventual recycling or degradation, or are packaged into secretory granules that then fuse with the plasma membrane, thereby releasing cytokines into the extracellular milieu (Figure 1). Under this classical secretory pathway, it is not surprising that a proportion of cytokines could be shunted into EVs, but it has been unclear how much of a given cytokine that is produced by a cell associates with EVs as compared to being freely secreted. A recent study identified that ~35% of the VEGF that is secreted by cancer cells is associated with small EVs (i.e., <200 nm in diameter) [53]. In another recent study, Fitzgerald and colleagues analyzed levels of 33 cytokines in eight different types of cultured cells, tissue explants and body fluids and, for each cytokine, compared the amounts that are EV-associated and freely secreted [51]. These authors found that all of the 33 cytokines can associate with EVs and that the degree of EV association of a given cytokine largely depends on the cell type or context [51]. For example, IL-6 and monocyte chemoattractant protein-1 were secreted by monocytes almost entirely in soluble form, but released exclusively in EV-associated form by T cells [51]. In this study, EVs were not fractionated and it is unclear whether cytokines vary in their distribution in different types of EVs. Notably, the authors found that activation of monocytes by lipopolysaccharides altered the secretion of nine cytokines from a predominantly EV-associated form to a soluble form [51]. These findings implicate that the distribution of cytokines between EV-associated and freely secreted forms can be altered by external cues.

### 3.2. Luminal versus Membrane Localization in EVs

Whereas earlier studies assumed that cytokines are encapsulated within EVs [9], recent studies have identified that several cytokines localize to the surface of EVs. In their analysis of 33 cytokines released by eight types of normal cells/tissues, Fitzgerald and colleagues identified that 18 cytokines are preferentially encapsulated in EVs, whereas IL-8 and chemokine (C-X-C motif) ligand 1 (CXCL1) predominantly localize to the surface of EVs [51]. IL-8 and CXCL1 also localize to the surface of cancer cell-derived EVs [53]. Other cytokines vary in their distribution between the lumen and membrane of EVs depending on the cell type [51]. Several variables appear to govern whether a cytokine is destined to be encapsulated within EVs or bound to the EV membrane. These variables include the mechanisms by which the cytokine is processed or secreted, the type of EV, and the type of membrane proteins that are displayed on the EV surface. An example of a cytokine that is predominantly directed into the luminal compartment of EVs is FGF-2. FGF-2 lacks a signal peptide and does not undergo classical ER-to-Golgi trafficking. Instead, FGF-2 is recruited to the inner leaflet of the plasma membrane where it is released from cells via encapsulation in ectosomes [54], or through pores in the membrane [55]. Because receptors for various cytokines are expressed on EV membranes [16,17], one mechanism by which a cytokine could associate with the EV surface is by binding its receptor on EVs. One example is TGF-β which associates with the surface of cancer cell-derived exosomes via the TGF-β type III receptor [56]. The temporality of this interaction is unclear. During endocytosis, cytokines are bound to their receptors (Figure 1), but it is unlikely that cytokines are sorted into exosomes along with their receptor because the acidic pH of late endosomes causes most receptor-ligand complexes to dissociate [57]. It might therefore be expected that a cytokine associates with its cognate receptor on the EV surface after both the cytokine and EVs have been secreted.

Intriguingly, recent studies of VEGFs indicate that the association of a cytokine with the EV surface might not occur as a post-secretion event. Several isoforms of VEGF are generated through alternative splicing of *VEGFA* mRNA [58]. Ko and colleagues identified that the 189 amino acid isoform of VEGF (VEGF_189_), but not other VEGF isoforms, is preferentially enriched in small EVs that are secreted by ovarian, colorectal, and renal cancer cells and are present in body fluids of patients with these cancers, and that VEGF_189_ is bound to the surface of these EVs [53]. In this study, small EVs were defined as EVs of <200 nm in diameter and predominantly comprised exosomes. By evaluating the amounts of VEGF_189_ in small EVs secreted by cancer cells that express VEGF_189,_ and assaying the binding of ‘free’ recombinant VEGF_189_ to small EVs secreted by VEGF-deficient cancer cells_,_ the authors estimated that post-secretion binding accounts for only ~25% of the VEGF_189_ in small EVs, implicating that de novo synthesized VEGF_189_ is largely sorted to small EVs [53]. Furthermore, the authors identified that VEGF_189_ interacts with the surface of small EVs through heparan sulfate binding (Figure 2). Of the commonly expressed isoforms of VEGF, VEGF_189_ has substantially higher affinity for heparan sulfate than VEGF_165,_ and VEGF_121_ does not bind heparan sulfate [59]. Because of the high affinity of VEGF_189_ for heparan sulfate and the abundance of HSPGs in small EV membranes, the selective sorting of VEGF_189_ to small EVs might be mediated by HSPGs. HSPGs might also mediate sorting to small EVs of several other cytokines that bind heparan sulfate, such as IL-8 and CXCL1 [60,61]. By contrast to the selective association of VEGF_189_ with small EVs, Feng and colleagues identified a 90 kDa form of VEGF (VEGF_90K_) on the surface of breast cancer cell-derived microvesicles of 0.5 to 1.0 μm in diameter [62]. VEGF_90K_ was generated by crosslinking of VEGF_165_ by tissue transglutaminase, and associated with microvesicles through binding heat shock protein 90 (Hsp90) [62] (Figure 2). VEGF_90K_ was not detected in small EVs and, conversely, other isoforms of VEGF were not detected in microvesicles [53,62]. Detailed tracing of VEGF_165_ trafficking has revealed that VEGF_165_ undergoes unconventional secretion whereby VEGF_165_ is retained on the outer surface of the plasma membrane and is then released on vesicles that shed from the cell surface [63]. Collectively, these findings implicate that different isoforms of VEGF are sorted by distinct mechanisms, resulting in localization to the membranes of different types of EVs (Figure 2).

## 4. Functional Significance of EV Membrane-Associated Proteins in Tumor Angiogenesis

Uptake of EVs by recipient cells has been thought to be obligatory for EV-mediated intercellular communication. This uptake can occur through clathrin-dependent endocytosis and/or clathrin-independent pathways such as caveolin-mediated endocytosis, macropinocytosis, phagocytosis, and lipid raft–mediated internalization [6,7]. EVs have been found to encapsulate various luminal cargo with angiogenic properties, and the activity of these constituents is contingent upon EV uptake by recipient cells [9,10,11,12,13]. However, increasing evidence indicates that the surfaces of EVs display a repertoire of membrane-bound proteins that stimulate various steps in the angiogenic cascade, and may do so independently of EV uptake. This section discusses how tumor angiogenesis is stimulated via paracrine signaling mediated by EV membrane-associated proteins, how signaling-competence of these proteins is modulated by their interactions with the EV membrane, and how the association of pro-angiogenic proteins with the EV membrane might confer selective advantages for tumor growth and metastasis.

### 4.1. Angiogenic Pathways Mediated by EV Integral Membrane Proteins

Degradation of the basement membrane and remodeling of the extracellular matrix (ECM) are essential to enable endothelial cells to invade tissues and form new vessels [1]. One of the earliest findings that implicated the significance of EV membrane-associated proteins in tumor angiogenesis was that both cancer cells and endothelial cells secrete EVs that display membrane-associated matrix metalloproteinases and other ECM-degrading proteases which retain their proteolytic activity [64,65,66]. Since that time, other types of membrane proteins in cancer cell-derived EVs have been shown to stimulate tumor angiogenesis by directly triggering signaling pathways in endothelial cells that stimulate survival, migration and/or tube forming ability of these cells. One example is EGFR. Microvesicles released by lung, colorectal and skin cancer cells have been shown to transfer oncogenic EGFR to endothelial cells and to elicit EGFR-dependent MAPK and AKT activation in these cells [67]. Microvesicle-associated EGFR triggered endothelial cells to express VEGF that in turn led to autocrine activation of VEGF receptor-2 (VEGFR2), which mediates the majority of the angiogenic effects of VEGF [67]. These EGFR-dependent responses were inhibited when uptake of microvesicles by endothelial cells was blocked, indicating that internalization of EV-associated EGFR is essential for its bioactivity [67].

By contrast, several other EV integral membrane proteins can elicit signaling by interacting with proteins on the plasma membrane of endothelial cells. Sato and colleagues recently identified that the ephrin type B receptor 2 (EPHB2) is present on small EVs secreted by head and neck cancer cells, and that EV-associated EPHB2 stimulates tumor angiogenesis by activating the STAT3 signaling pathway via engagement of ephrin-B2 on the surface of endothelial cells [68]. Whereas the cell adhesion molecule E-cadherin is expressed as a full-length protein on the plasma membrane of ovarian cancer cells [69], Tang and colleagues identified that the surface of ovarian cancer cell-derived exosomes contain the soluble ectodomain of E-cadherin (sE-cad) [70]. Although the mechanism by which sE-cad interacts with the exosomal surface is unclear, sE-cad was found to stimulate tumor angiogenesis by forming a heterodimer with VE-cadherin on the surface of endothelial cells which in turn activated β-catenin and NF-κB signaling [70]. Endothelial cells and several other types of stromal cells also secrete EVs that contain membrane proteins with pro-angiogenic properties. For example, Sheldon and colleagues identified that endothelial cells secrete exosomes that contain the membrane-anchored Notch ligand Delta-like 4 (Dll4) [71]. These authors found that Dll4 is transferred by exosomes to the plasma membrane of recipient endothelial cells, which in turn inhibited Notch signaling and increased vessel branching [71]. Platelet-derived microvesicles have been shown to transfer CD41 (as known as integrin α-2b) to the surface of lung cancer cells, which in turn induced expression of pro-angiogenic cytokines such as VEGF, IL-8, and hepatocyte growth factor (HGF) [72].

Given that many of the pro-angiogenic integral membrane proteins on EVs retain signaling-competence, these proteins likely retain the same orientation on the EV membrane as on the plasma membrane of the parental cell. This assumption could particularly hold for ectosome-associated membrane proteins because ectosomes form by outward budding of the plasma membrane (Figure 1). The MVB pathway involves two invagination steps: firstly, inward budding of the plasma membrane during endocytosis and, secondly, budding of the endosomal membrane towards the endosomal lumen to form ILVs that are then released as exosomes (Figure 1). Under this two-step invagination model, membrane proteins in exosomes are expected to have the same orientation as on the plasma membrane. However, a study by Cvjetkovic and colleagues revealed that this may not always be the case. By using a multiple proteomics approach that combined proteinase treatment and biotin tagging, these authors identified that a striking proportion of proteins are displayed in EVs in a topologically reversed orientation to their plasma membrane counterparts [73]. Of the 49 transmembrane and lipid-anchored proteins that the authors characterized, 32 of these proteins retained conventional topology in EVs, whereas 16 proteins were ‘inside-out’ [73]. Because EVs were not fractionated in this study, it is not clear whether the prevalence of reversed topology of proteins varies between ectosomes and exosomes. However, the findings of Cvjetkovic and colleagues have several important implications. If topologically altered, membrane proteins in EVs might elicit signaling pathways that are different to those evoked by their plasma membrane counterparts. Furthermore, the conformation of proteins could be altered by their interaction with the EV membrane. As a result of this conformational change, EV membrane-associated proteins might not be recognized by therapeutic antibodies, as described later in Section 5.1.

### 4.2. Angiogenic Pathways Mediated by EV Membrane-Associated Ligands

In their analysis of protein topology in EVs, Cvjetkovic and colleagues not only identified that many integral membrane proteins are oriented ‘inside-out’ in EVs but also that many cytosolic proteins localize on the surface of EVs [73]. As described in Section 3.2, EVs that are secreted by cancer cells and by stromal cells display various cytokines on the vesicular surface. Several of these EV membrane-associated cytokines, such as VEGF, IL-8, CXCL1, and TGF-β, have well-established functions in stimulating endothelial cell growth, migration, and/or tube formation [1,58], and have been shown to be signaling-competent. Microvesicle-associated VEGF_90K_ has been shown to stimulate VEGFR2 phosphorylation in endothelial cells and tube formation [62]. Small EV-associated VEGF_189_ has also been shown to stimulate VEGFR2 phosphorylation, endothelial cell migration, and tube formation in vitro and to increase tumor microvessel density in vivo [53]. The ability of small EVs carrying VEGF_189_ to stimulate endothelial cell migration and tube formation was not inhibited when EV uptake by endothelial cells was blocked, but was abrogated when endothelial cells were treated with an antibody that blocks ligand binding to VEGFR2 [53]. These findings indicated that small EV-associated VEGF_189_ interacts with the extracellular domain of VEGFR2 and elicits VEGFR2 signaling in endothelial cells independently of uptake. VEGF isoforms are biologically active as homodimers [74] and VEGF_189_ has been found to be present on small EVs as a homodimer [53]. By contrast, TGF-β localizes on the surface of exosomes of cancer cells and mast cells in predominantly latent form [56,75]. These exosomes have been shown to activate TGF-β-dependent Smad2/Smad3 signaling in recipient cells, although the mechanism by which exosomal TGF-β is converted to an active form is unclear [56,75].

### 4.3. Alterations in Stability and Signaling of EV Membrane-Associated Proteins

The secretion of EVs was initially described as a means of eliminating unneeded biomolecules from cells [76]. It is now recognized that EVs are not merely waste carriers, but can be harnessed by tumors to support their growth. It has been thought that a fundamental advantage conferred by EVs is the protection of informational cargo from degradation [5,9]. This may be the case for cargo that is encapsulated in EVs, but it might be expected that proteins on the surface of EVs are not protected. Most freely secreted cytokines have short half-lives in the circulation (typically <1 h) [77]. Notably, it has been shown that levels of VEGF_189_ on the surface of cancer cell-derived small EVs remain almost unchanged at 24 h following incubation in plasma at 37 °C, indicating that the association of this pro-angiogenic cytokine with the EV surface profoundly increases its stability [53]. It is also possible that the association of a protein with the EV membrane changes its conformation and thereby its potency. It has been found that microvesicle membrane-associated VEGF_90K,_ which comprises crosslinked VEGF_165_, is two-fold more effective in stimulating endothelial tube formation than an equivalent amount of free VEGF_165_ [62]. Amphiregulin, an EGFR ligand, is present on colorectal cancer cell-derived exosomes, and exosomal amphiregulin has been shown to be five-fold more effective in stimulating cell invasiveness than the soluble ligand [78]. Exosome membrane-associated TGF-β has been found to elicit sustained, low amplitude signaling that evokes greater functional effects than the transient, high amplitude signaling elicited by soluble TGF-β [75]. In addition to enhanced stability, EV membrane-associated cytokines and other proteins might confer stronger advantages to tumors, through qualitative and quantitative differences in signaling, than their soluble or cellular counterparts. Furthermore, angiogenesis might be fostered more efficiently by factors that are delivered in a surface-concentrated form by EVs than by diffuse, soluble factors. Intriguingly, as compared to EV-associated cytokines released by cultured cells and tissue explants, EV-associated cytokines in plasma have been found to be predominantly displayed on the EV surface than encapsulated in EVs [51]. Through conveyance on circulating EVs and with the potential to efficiently elicit signals without the need for EV uptake, pro-angiogenic EV membrane-associated cytokines might mediate long-range signaling that is particularly conducive for metastasis.

## 5. Clinical Significance of EV Membrane-Associated Proteins

Agents that inhibit angiogenesis have been among the most extensively studied anti-cancer therapies. Bevacizumab, a humanized antibody that neutralizes VEGF, was the first anti-angiogenic agent approved for the treatment of solid tumors and remains widely used [79]. A more recently approved therapeutic antibody is ramucirumab which targets the ligand-binding domain of VEGFR2 [80]. Other approved agents include multi-kinase inhibitors, such as sorafenib and sunitinib, which target the VEGF receptor (VEGFR), PDGF receptor (PDGFR), and KIT receptor tyrosine kinases [81]. A substantial limitation of these anti-angiogenic agents is that they often result in transient clinical benefit, followed by a restoration of tumor growth and progression [82]. This section discusses recent insights into the mechanisms by which EV proteins, and membrane-associated proteins in particular, contribute to resistance to anti-angiogenic therapy. In addition, the potential of exploiting EVs to improve clinical outcomes will be discussed.

### 5.1. Significance of EV Proteins in Resistance to Anti-Angiogenic Therapy

The resistance of tumors to anti-angiogenic therapy has been attributed to several mechanisms such as the utilization by tumors of existing vessels [83], infiltration of pro-angiogenic myeloid cells [84] and treatment-induced intratumoral hypoxia stemming from excessive blood vessel pruning [85]. Hypoxia induces both quantitative and qualitative changes in EV-mediated signaling. Hypoxia stimulates cells to secrete higher numbers of exosomes [86] and to shed microvesicles from the plasma membrane [87]. Furthermore, hypoxia triggers dynamic changes in transcription which are largely orchestrated by hypoxia-inducible factor-1, resulting in increased levels of pro-angiogenic factors such as VEGF and members of the FGF, angiopoietin, and ephrin families [88]. Hypoxic glioblastoma cells have been found, as a result of hypoxia-induced transcriptional reprogramming, to secrete EVs that are enriched in pro-angiogenic cytokines and are more effective than EVs secreted by normoxic glioblastoma cells in stimulating tumor growth, vessel density, and pericyte vessel coverage [49]. Studies of liver cancer models have shown that tyrosine kinase inhibitor (TKI) treatment induces the release of VEGF-enriched exosomes that in turn stimulate angiogenesis [89], and that EVs can also enable evasion from TKI-induced cell death by activating VEGF-independent pathways such those mediated by HGF [90]. Collectively, these findings suggest that the efficacy of anti-angiogenic therapy could be reduced through hypoxia-induced increases in EV release and in the content of pro-angiogenic cytokines in EVs. Paracrine signaling can be also mediated by membrane-associated proteins that are enriched in EVs secreted by hypoxic cells. It has been shown that microvesicles released by hypoxic glioblastoma cells contain high amounts of tissue factor that stimulates tumor angiogenesis by cleavage activation of protease-activated receptor 2 on endothelial cells [91]. Furthermore, exosomes secreted by hypoxic colorectal cancer cells have been found to be enriched in Wnt4 that stimulates endothelial cell proliferation and migration by inducing β-catenin signaling [92].

In addition to treatment-induced hypoxia and resultant changes in EV release and composition, there are several other mechanisms by which EVs could decrease efficacy of anti-angiogenic therapy. On one hand, EVs might mask the therapeutic agent. Simon and colleagues identified that bevacizumab is internalized by glioblastoma cells, trafficked into endosomes and then displayed on EVs secreted by these cells [93]. This EV-associated bevacizumab was not able to bind VEGF [93]. These findings suggest that decreased efficacy of bevacizumab might stem from its sequestration in EVs. On the other hand, EVs might mask the therapeutic target. Bevacizumab is thought to neutralize all isoforms of VEGF, but previous studies have mostly characterized its binding to soluble VEGF [94]. Ko and colleagues recently showed that small EV-associated VEGF_189_ is signaling-competent but is not neutralized by bevacizumab [53]. The authors found that VEGF_189_ associates with the surface of small EVs through heparin-binding, and that the engagement of VEGF_189_ with high molecular weight (HMW) heparin reduces its recognition by bevacizumab [53]. It has been reported that residues in the β5- and β6- sheets and intervening loop of VEGF are critical for forming a high-affinity complex with bevacizumab [94], and that the β-sheet content of VEGF is diminished when VEGF is bound to HMW heparin [95]. The inability of bevacizumab to neutralize small EV-associated VEGF_189_ might therefore stem from a conformational change in VEGF_189_ that is induced by its engagement with heparin (Figure 2). In another study, Feng and colleagues similarly identified that microvesicle-associated VEGF_90K_ activates VEGFR signaling and is not neutralized by bevacizumab [62]. The authors found that VEGF_90K_ associates with microvesicles through interacting with Hsp90, and that treatment with an Hsp90 inhibitor releases VEGF_90K_ from microvesicles, restoring sensitivity of the ligand to bevacizumab (Figure 2). Taken together, these studies implicate that EVs could decrease the efficacy of bevacizumab by rendering VEGF unrecognizable to the therapeutic agent.

### 5.2. Exploiting EVs to Improve Clinical Outcomes

How can biological studies of EVs be translated into studies that improve outcomes of cancer patients? One major limitation of anti-angiogenic therapy has been the lack of reliable biomarkers that can predict clinical response [82]. Findings of several independent studies of the predictive value of total circulating VEGF have been discordant. Baseline plasma levels of total VEGF have been found to be predictive of bevacizumab treatment benefit in some studies [96,97], but not in others [98,99]. Insensitivity of patient-derived breast cancer xenografts to bevacizumab has been found to be associated with high levels of secreted microvesicle-associated VEGF_90K_ [62]. Furthermore, baseline plasma levels of small EV-associated VEGF_189_ have been found to be approximately five-fold higher in patients with progressing disease than in those with stable or regressing disease in a cohort of patients with newly diagnosed metastatic renal cell carcinoma who were treated pre-surgically with single-agent bevacizumab [53]. However, in this cohort, there was no significant difference in baseline levels of total VEGF between patients who had progressing disease and those who had stable or regressing disease [53]. These findings raise the possibility that baseline levels of EV-associated VEGF might be more informative for assessing bevacizumab treatment benefit than levels of total VEGF (Figure 2).

In addition to the potential use of EV-associated proteins as biomarkers, the possibility of using natural EVs as therapeutic agents has been raised by findings that EVs secreted by some types of normal cells can inhibit angiogenesis. The heart is renowned for its immunity to cancer, and EVs isolated from heart progenitor cells have been shown to inhibit tumor growth and angiogenesis in a fibrosarcoma model [100]. Microparticles derived from apoptotic T cells have also been shown to decrease lung tumor growth and angiogenesis [101]. Moreover, the use of EVs as vehicles to deliver therapeutic agents to cancer cells has attracted substantial interest [102]. Liposomes are clinically used to deliver anti-cancer drugs, but have several limitations such as their ability to trigger innate immune responses and to be phagocytosed [103]. EVs are advantageous as delivery vehicles by virtue of their natural surface proteins. The use of autologous EVs can minimize induction of immune responses, and the membrane protein CD47 on EVs confers protection from phagocytosis [102]. Furthermore, EVs are able to cross the blood–brain barrier [104]. Although the uptake of EVs by endocytosis is problematic for drug delivery, a recent study circumvented this limitation by conjugating a cell-penetrating peptide to the membrane of EVs which facilitated uptake by macropinocytosis and delivery of therapeutic cargo into the cytosol [105]. EV membrane proteins have also been modified to facilitate cell-specific delivery. EVs can exhibit intrinsic tropism depending on their membrane protein composition. For example, it has been shown that exosomes expressing integrin αvβ5 bind to Kupffer cells, whereas exosomes that express integrins α6β4 and α6β1 home to the lung [106]. However, systemically delivered exosomes mainly accumulate in the liver, gastrointestinal tract, lung, kidney, and spleen [107,108]. To target EVs to tumors, Ohno and colleagues modified embryonic kidney donor cells to express GE11, a peptide that binds to EGFR but is less mitogenic than EGF [109]. Exosomes of these donor cells expressed GE11 on their surface and could deliver an incorporated therapeutic microRNA to EGFR-positive breast tumor xenografts [109]. EVs have also been modified to display membrane proteins that facilitate targeting to endothelial cells. Wang and colleagues modified exosomes to display an Arg-Gly-Asp peptide that binds integrin αvβ3 on endothelial cells, and demonstrated that this modification enhances targeting of exosomes to the vasculature [110]. Although the pharmacokinetic and pharmacodynamic profiles of therapeutic EVs require further investigation, these recent studies collectively indicate that therapeutic EVs could be a promising class of agents for anti-angiogenic therapy.

## 6. Conclusions

In summary, there is increasing evidence that EVs released by cancer cells and by stromal cells display a repertoire of integral membrane proteins and membrane-anchored proteins that stimulate tumor angiogenesis and can enable tumors to evade anti-angiogenic therapy. However, there remain several gaps-in-knowledge. Firstly, functional studies to date have focused on a given type of EV, and differences in paracrine signaling mediated by different types of EVs are poorly understood. More comparative analysis of the composition and functional significance of membrane proteins between different types of EVs is needed. Paramount to these efforts is the development of more rigorous methods to isolate and clearly distinguish the different types of EVs. Secondly, further investigation is needed into the mechanisms by which pro-angiogenic proteins interact with the surface of EVs, and how the interactions of these proteins with the EV membrane modulate their topology, signaling competence and stability. As discussed, recent studies of EV membrane-associated VEGF highlight the importance of evaluating differences in the conformation of EV-associated proteins and their cellular or soluble counterparts in designing therapeutic agents that target these proteins. Thirdly, recent studies indicate that EVs are a promising class of therapeutic vehicles. Further investigation of EV membrane proteins will yield important insights for facilitating cell-specific delivery and the targeting of specific subcellular destinations.

## Figures and Tables

**Figure 1 ijms-21-05418-f001:**
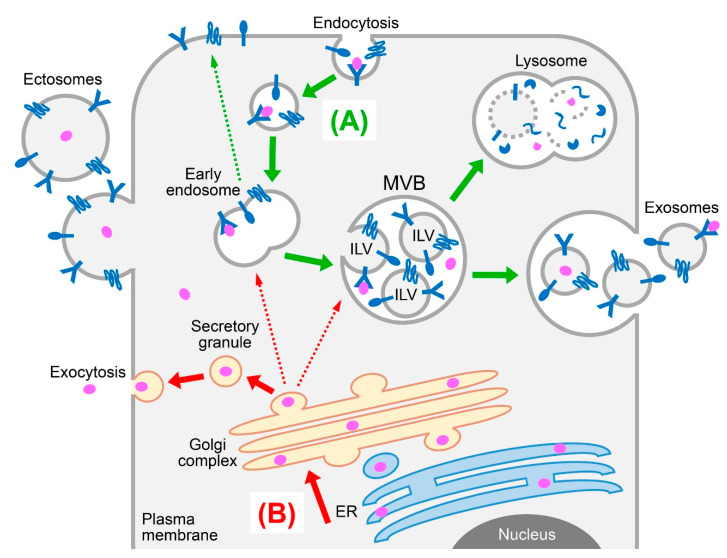
Schematic representation of protein sorting and secretory pathways. (**A**) Cell surface receptors and other integral membrane proteins (shown in dark blue) are internalized by endocytosis and routed to early endosomes (green arrow). From there, the proteins are recycled back to the plasma membrane, or are sorted into intraluminal vesicles (ILV) that form through inward budding of endosomes and give rise to multivesicular bodies (MVB). Fusion of MVBs with lysosomes results in degradation of ILVs and their cargo, whereas fusion of MVBs with the plasma membrane results in release of ILVs as exosomes. Unlike exosomes, ectosomes (also known as microvesicles or microparticles) form through outward budding of the plasma membrane. (**B**) Most receptor ligands (shown in pink) are trafficked to the endoplasmic reticulum (ER) and then to the Golgi complex (red arrow). From there, the proteins are routed to endosomes, or are packaged into secretory granules that then fuse with the plasma membrane, resulting in release of proteins into the extracellular milieu.

**Figure 2 ijms-21-05418-f002:**
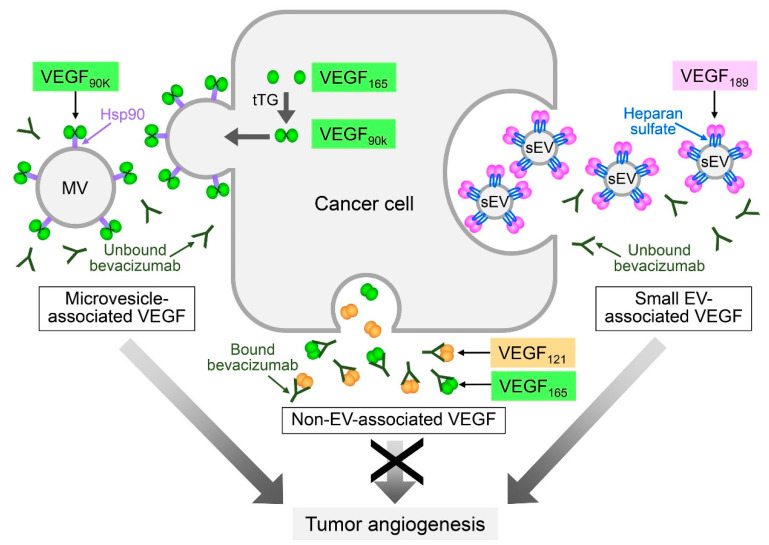
Differential sorting of vascular endothelial growth factor (VEGF) isoforms to EVs and impact on anti-angiogenic therapy. Alternative splicing of vascular endothelial growth factor A *(VEGFA)* mRNA yields several VEGF isoforms of which the 121, 165, and 189 amino acid variants are the most commonly expressed isoforms in tumors. The VEGF_189_ isoform is selectively enriched in cancer cell-derived small EVs (sEV). VEGF_189_ is present on the surface of small EVs as a covalent homodimer and interacts with the EV surface through binding heparan sulfate. By contrast, VEGF_90K_ is selectively enriched in cancer cell-derived microvesicles (MV). VEGF_90K_ comprises VEGF_165_ that is crosslinked by tissue transglutaminase (tTG) and interacts with the EV surface through binding heat shock protein 90 (Hsp90). Both small EV-associated VEGF and microvesicle-associated VEGF stimulate VEGF receptor signaling and tumor angiogenesis, but are not neutralized by the therapeutic VEGF antibody bevacizumab.

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
