# Peer review of "Extracellular Vesicle Membrane-Associated Proteins: Emerging Roles in Tumor Angiogenesis and Anti-Angiogenesis Therapy Resistance"

_ijms, 2020, doi:10.3390/ijms21155418_

Round 1

Reviewer 1 Report

In the present review, the Authors very well summarize the present knowledge about extracellular vesicle involvement in the promotion of angiogenesis in tumors.

They clearly introduce what are EVs and how heterogeneous they are, selecting the cutting edge references about their classification. Next, the molecular mechanisms at the basis of EV biogenesis and how membrane-associated and cargo molecules are sorted into the vesicles are extensively described.

Then the main topic of the work is addressed; the role of EV membrane-associated proteins in tumor angiogenesis. The review is focused on the importance of both the integral membrane proteins (e.g.: EGFR) and the membrane-associated ligands (such as VEGF, IL-8, CXCL1 and TGF-β), highlighting their impact in the resistance to anti-angiogenic therapies. Indeed, the Authors stress the point that EVs should be exploited both as biomarkers to evaluate the drug-resistance profile of the specific tumor, or, in future, as therapeutic agents.

Broad comments:

The review is very well written and the text is pleasantly fluid.

The topic is broad and in constant evolution. The scientific community is very active in the extracellular vesicle field and, for this reason, it is not surprising that just few weeks ago (2020 Jun 22) a review has been published about EVs and angiogenesis: “Tumor cells derived-exosomes as angiogenenic agents: possible therapeutic implications.” by Ahmadi M, Rezaie J. on the J Transl Med.

For completeness, I suggest the Authors to include a citation of this newly published work in their review.

Even if the two reviews naturally overlap in part of the background information delivered, the work by Ko and Naora is a very important contribution to the current picture of the EV-membrane proteins, that are getting more and more attention in the clarification of angiogenesis regulation.

For this reason, I recommend the publication of this review, that will be of great interest for the scientific community interested non only in the in angiogenesis/tumor field, but also in the broad EV research field.

Specific comments:

In Figure1, I encourage the Authors to better highlight the letters (A) and (B), e.g. by increasing the font size, putting a white box on the background, etc. since I found a little difficult to locate them at first sight.

Author Response

We thank the reviewer for his/her positive comments and constructive suggestions. As recommended by Reviewer 1:

  1. We have included citation of the recent review article by Ahmadi & Rezaie (2020) J Transl Med (Cited as Reference 4 on page 1, line 29 and included in Reference section, page 14, line 550).
  2. We have revised Figure 1 (page 11) to improve clarity by increasing the font size of the letters (A) and (B) and by placing a white box behind these letters.

Reviewer 2 Report

In this manuscript, Song Yi Ko and Honami Naora summarized the biology of extracellular vesicle (EV) membrane-associated proteins. EVs contain both transmembrane proteins and secretory proteins. Vascular endothelial growth factor (VEGF) is one of secretory proteins, but there are microvesicle- and small EV-mediated and non-EV-mediated secretion, and the form of secretion is very important for the treatment of cancer. This review provides an easy-to-understand explanation of EV membrane associated VEGF in cancer. Overall, this manuscript is well organized and well written. Thus, I think the manuscript meet the criteria of International Journal of Molecular Sciences.

Author Response

We thank the reviewer for his/her positive comments. No revisions were requested by Reviewer 2.